# Control of ciliary transcriptional programs during spermatogenesis by antagonistic transcription factors

**Weihua Wang[1,2]\*[†], Junqiao Xing[1,2†], Xiqi Zhang[1†], Hongni Liu[1,2], Xingyu Liu[1,3], Haochen Jiang[1,2], Cheng Xu[1,2], Xue Zhao[1,2], Zhangfeng Hu[1,2,4]\***

[1]Institute of Microalgae Synthetic Biology and Green Manufacturing, School of Life Sciences, Jianghan University, Wuhan, China; [2]Hubei Engineering Research Center for Protection and Utilization of Special Biological Resources in the Hanjiang River Basin, School of Life Sciences, Jianghan University, Wuhan, China; [3]No.1 Middle School Affiliated to Central China Normal University, Wuhan, China; [4]Hubei Key Laboratory of Environmental and Health Effects of Persistent Toxic Substances, Jianghan University, Wuhan, China

**\*For correspondence:**
wangwh2019@163.com (WW);
huzhangfenghzf@163.com (ZH)

[†]These authors contributed equally to this work

## eLife Assessment

This **valuable** study presents data suggesting the critical roles of two ancient proteins, XAP5 and XAP5L, in regulating the transcriptional program of ciliogenesis during mouse spermatogenesis. The supporting data are **solid**, and this work will be of interest to biomedical researchers studying ciliogenesis and reproduction.

**Abstract** Existence of cilia in the last eukaryotic common ancestor raises a fundamental question in biology: how the transcriptional regulation of ciliogenesis has evolved? One conceptual answer to this question is by an ancient transcription factor regulating ciliary gene expression in both uni- and multicellular organisms, but examples of such transcription factors in eukaryotes are lacking. Previously, we showed that an ancient transcription factor X chromosome-associated protein 5 (Xap5) is required for flagellar assembly in *Chlamydomonas*. Here, we show that Xap5 and Xap5-like (Xap5l) are two conserved pairs of antagonistic transcription regulators that control ciliary transcriptional programs during spermatogenesis. Male mice lacking either Xap5 or Xap5l display infertility, as a result of meiotic prophase arrest and sperm flagella malformation, respectively. Mechanistically, Xap5 positively regulates the ciliary gene expression by activating the key regulators including Foxj1 and Rfx families during the early stage of spermatogenesis. In contrast, Xap5l negatively regulates the expression of ciliary genes via repressing these ciliary transcription factors during the spermiogenesis stage. Our results provide new insights into the mechanisms by which temporal and spatial transcription regulators are coordinated to control ciliary transcriptional programs during spermatogenesis.

## Introduction

Cilia and flagella are evolutionarily conserved hair-like microtubule organelles that project from the surfaces of most eukaryotic cells (*Derderian et al., 2023*; *Mitchell, 2017*). They play essential roles in sensory reception, signal transduction, and cell movement (*Goetz and Anderson, 2010*; *Nachury and Mick, 2019*; *Singla and Reiter, 2006*). The dysfunction of cilia in humans leads to an emerging

class of genetic diseases called ciliopathies classified into first- and second-order ciliopathies (*Ishikawa and Marshall, 2011*; *Lovera and Lüders, 2021*; *Reiter and Leroux, 2017*). First-order ciliopathies are caused by aberrations in ciliary proteins, and second-order ciliopathies occur due to defects in non-ciliary proteins, such as transcription factors, that are required for cilia formation and function (*Reiter and Leroux, 2017*).

Cilia are complex organelles and are dynamically regulated during development and the cell cycle (*Eggenschwiler and Anderson, 2007*; *Kasahara and Inagaki, 2021*; *Santos and Reiter, 2008*). The transcriptional regulation of ciliary genes must be precisely coordinated during ciliogenesis (*Choksi et al., 2014*; *Collins et al., 2021*; *Lewis and Stracker, 2021*; *Thomas et al., 2010*). In metazoans, several transcription factors, including Foxj1 and Rfxs, have been shown to be involved in directing the expression of ciliary genes (*Stubbs et al., 2008*; *Swoboda et al., 2000*; *Yu et al., 2008*). These key ciliary transcription factors show dynamic expression patterns during development and differentiation (*Stubbs et al., 2008*; *Swoboda et al., 2000*; *Yu et al., 2008*). However, how these key transcriptional programs are spatially and temporally controlled by cell type-specific transcription factors and signaling pathways in order to activate specific target ciliary genes and generate diverse cilia is largely unknown.

Despite transcriptional regulation of ciliary genes is required for ciliogenesis in both uni- and multicellular organisms, the underlying mechanism mediating ciliary gene expression may be fundamentally different, as the two major transcription factors, Foxj1 and Rfxs, are absent from many unicellular organisms (*Chu et al., 2010*; *Li et al., 2018*; *Piasecki et al., 2010*; *Vij et al., 2012*). Cilia and ciliary genes have been highly conserved throughout evolution, suggesting that the regulation of ciliary genes could be programmed by yet undiscovered transcriptional mechanisms, which possibly coevolved with multicellularity. Recently, an ancient transcription factor, X chromosome-associated protein 5 (Xap5), is found to regulate ciliary gene expression in a unicellular organism, *Chlamydomonas reinhardtii* (*Li et al., 2018*). Xap5 is highly conserved among different species and is widely expressed across tissues (*Martin-Tryon and Harmer, 2008*; *Mazzarella et al., 1997*). In vertebrates, *Xap5-like* (*Xap5l*) with an intronless open reading frame originated in Therians via retrotransposition of the ancestral *Xap5*, and it is highly expressed during spermatogenesis (*Sedlacek et al., 1999*; *Zhang et al., 2011*). However, whether Xap5 and Xap5l play essential roles in cilia development in multicellular organisms remains unknown.

Here, we show that Xap5/Xap5l are antagonistic transcription factors required for ciliary transcriptional programs during spermatogenesis. Xap5 is widely expressed in different tissues, while Xap5l is exclusively present in the testes. To explore the roles of Xap5/Xap5l in ciliogenesis, we generated a *Xap5l* knockout (KO) mouse line and found that *Xap5l* KO male mice are infertile due to the malformation of sperm flagella. Interestingly, male mice with the loss of Xap5 protein in germ cells also exhibited infertile, due to the arrest of spermatogenesis in the meiotic prophase stage. These data suggest that Xap5/Xap5l play crucial roles in sperm flagellar assembly in a spatiotemporal manner. Mechanistically, Xap5 positively modulates transcriptional rewiring of ciliary genes via activating the key regulators including Foxj1 and Rfxs during the early stage of spermatogenesis. Conversely, Xap5l negatively regulates the transcriptional program controlling ciliogenesis by repressing these ciliary transcription factors during the spermiogenesis stage. Thus, ciliary transcriptional programs are spatially and temporally controlled by Xap5/Xap5l antagonistic transcription factors during spermatogenesis.

## Results

### Xap5 and Xap5l are indispensable for male fertility

To explore the potential functions of Xap5/Xap5l during ciliogenesis in vivo, we performed western blot analysis to investigate the spatiotemporal expression of Xap5/Xap5l in various mouse tissues. We observed that Xap5 protein was widely expressed in diverse tissues and present at all stages of postnatal testes maturation, whereas Xap5l protein was mainly present in the testes and its level increased dramatically from postnatal day 21 (P21) and continued into adulthood (*Figure 1A, B*). Next, we identified the specific testicular cell types expressing *Xap5/Xap5l* using published single-cell RNA-seq data (*Jung et al., 2019*). *Xap5* mRNA was found predominantly expressed in spermatogonia, while *Xap5l* mRNA was observed in pachytene spermatocytes, and remained until elongating spermatids (*Figure 1C*). Consistent with the mRNA localization patterns, immunofluorescence staining showed

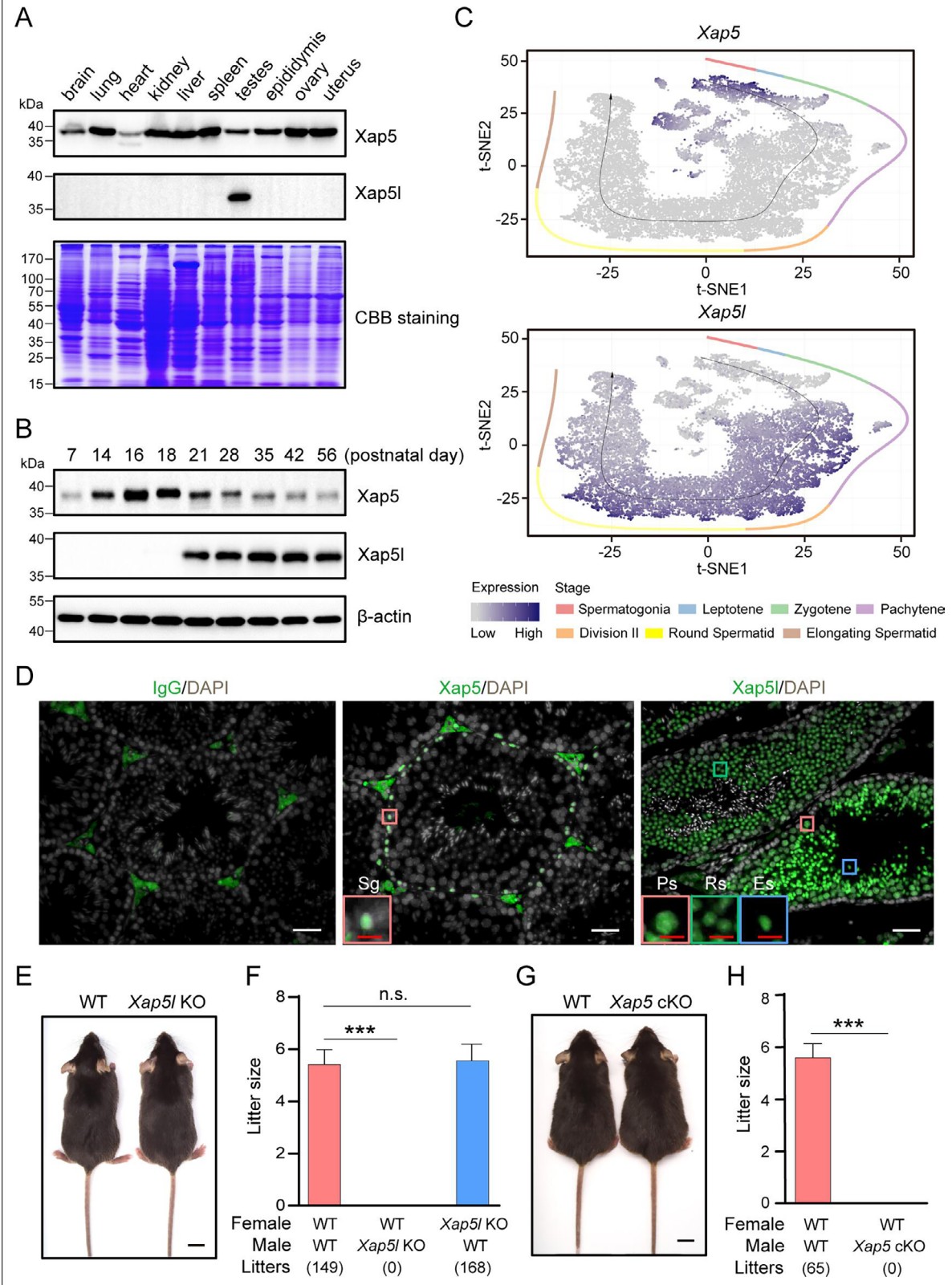

**Figure 1.** Loss of Xap5 and Xap5l proteins results in male mice infertility. (**A**) Detection of Xap5/Xap5l protein in different tissues of adult mice by western blot analysis. (**B**) Expression of Xap5/Xap5l protein during postnatal testicular development. β-Actin was used as control. (**C**) t-SNE plots displaying *Xap5/Xap5l* gene expression levels in individual cell types of mouse testis visualized using published single-cell data (*Jung et al., 2019*). Black arrow represents the developmental pseudotime corresponding to the developmental ordering of each cell progressing through

*Figure 1 continued on next page*

*Figure 1 continued*

spermatogenesis. Dot color intensity represents the expression level. Cells from different stages are color-coded. (**D**) Immunofluorescence staining of Xap5/Xap5l in adult WT testis. Representative spermatogonium (Sg), round spermatid (Rs), and elongating spermatid (Es) are indicated. Scale bars: 20 μm (white), 5 μm (red). (**E**) Similar development between WT and *Xap5l* KO mice. Scale bar: 1 cm. (**F**) Average litter size of pups obtained by fertility test. Each male was caged with two females for 2 months, five males were used for each genotype. ***p < 0.001, n.s. stands for not significant. (**G**) Similar appearance of WT and *Xap5* cKO mice. Scale bar: 1 cm. (**H**) Average litter size of pups obtained by fertility test. Each male was caged with two WT females for 2 months, four males were used for each genotype. ***p < 0.001. Error bars depict means ± SEM. All p-values were calculated using an unpaired, two-tailed Student's *t*-test.

The online version of this article includes the following source data and figure supplement(s) for figure 1:

**Source data 1.** Original files for western blot analysis displayed in *Figure 1A and B*.

**Source data 2.** File containing the original SDS-PAGE gel and western blot images for *Figure 1A and B*, indicating the relevant bands and corresponding samples.

**Figure supplement 1.** Localization of Xap5/Xap5l in testicular cells.

**Figure supplement 1—source data 1.** Original files for western blot analysis displayed in *Figure 1—figure supplement 1A–C*.

**Figure supplement 1—source data 2.** File containing the original SDS-PAGE gel and western blots for *Figure 1—figure supplement 1A–C*, indicating the relevant bands and corresponding samples.

**Figure supplement 2.** Generation of *Xap5l* KO and *Xap5* cKO mice.

**Figure supplement 2—source data 1.** Original files for genotyping and western blot analysis displayed in *Figure 1—figure supplement 2B, C, E and F*.

**Figure supplement 2—source data 2.** File containing the original agarose gel and western blot images for *Figure 1—figure supplement 2B, C, E, and F*, indicating the relevant bands and corresponding samples.

that Xap5 protein was mainly detected in the nuclei of spermatogonia, whereas Xap5l protein was present in the nuclei of pachytene spermatocytes and spermatids within the seminiferous tubules, and neither of Xap5 or Xap5l was detected in mature sperm (*Figure 1D*, *Figure 1—figure supplement 1*).

To investigate the physiological function of the uncharacterized Xap5/Xap5l protein, we attempted to generate global KO mice using the CRISPR-Cas9 system. We successfully established a *Xap5l* KO mouse line (*Figure 1—figure supplement 2A–C*) and found that while all *Xap5l* KO mice survived to adulthood and appeared indistinguishable from the wildtype (WT) mice, *Xap5l* KO males were sterile (*Figure 1E, F*). Notably, generating mosaic F0 founders with *Xap5* mutant allele proved challenging, and no F1 generation inherited the mutant alleles from the few F0 founders in over a year of breeding experiments, underscoring the critical roles of Xap5 in mouse development. Subsequently, we generated floxed-*Xap5* mice and crossed them with *Stra8-GFPCre* knockin mice (*Lin et al., 2017*) to produce *Xap5* germline-specific KO mice (*Xap5* cKO) (*Figure 1—figure supplement 2D–F*). Similar in appearance to controls (*Xap5^{fl/Y}*), all *Xap5* cKO males were also found to be sterile (*Figure 1G, H*). Collectively, these results suggest that Xap5/Xap5l play essential roles in spermatogenesis.

## Xap5 and Xap5l function at different stages of spermatogenesis

To uncover the cause of male infertility, we conducted gross and histological analyses of the testes from WT and *Xap5l* KO mice. There were no differences in the gross appearance of testis or in the weight of mouse body and testis between WT and *Xap5l* KO mice (*Figure 2A*), but the number of sperm collected from cauda epididymis of *Xap5l* KO mice was significantly lower than that from WT mice (*Figure 2B*), and sperm motility was significantly reduced in KO mice (*Figure 2C* and *Figure 2—videos 1 and 2*). Histological analysis did not reveal any severe abnormity in the seminiferous epithelial cycle of *Xap5l* KO testes (*Figure 2D*). Further examination of the morphology and structure of cauda epididymal sperm in *Xap5l* KO mice revealed significantly higher rates of abnormal flagella (*Figure 2E–G*), indicating that *Xap5l* is crucial for flagellar assembly during spermiogenesis.

We also investigated the spermatogenic defects in the seminiferous tubules and epididymis of *Xap5* cKO male mice. Notably, the testes of *Xap5* cKO males were significantly smaller compared to WT controls at P16 and older (*Figure 3A*). Many seminiferous tubules in P16 and older *Xap5* cKO males were vacuolated due to the absence of spermatocytes and spermatids, as indicated by the lack of XY body formation and the absence of PNA signal, a marker of spermatids and sperm, in *Xap5* cKO germ cells (*Figure 3B–D*). As a consequence, no sperm were observed in the adult cauda epididymides of *Xap5* cKO mice (*Figure 3E*). Taken together, these data indicate that germ cell development

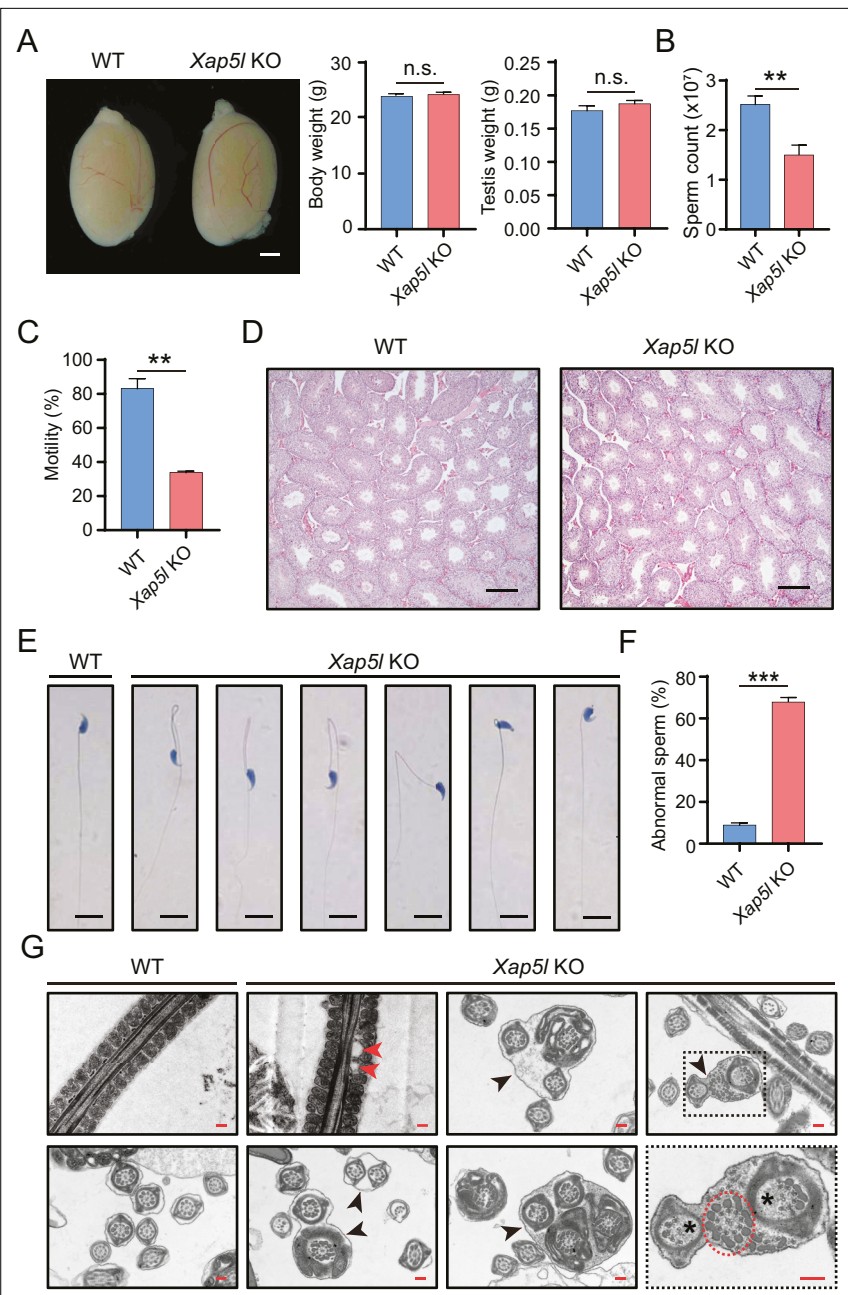

**Figure 2.** Xap5l is required for normal sperm formation. (**A**) Similar gross testes morphology, body weight, and testis weight between WT and *Xap5l* KO mice. Scale bar: 1 mm. *n* = 6, n.s. stands for not significant. (**B**) Significantly reduced caudal epididymal sperm counts in Xap5l KO mice compared with WT males. *n* = 5, **p < 0.01. (**C**) Significantly reduced sperm total motility of *Xap5l* KO mice. *n* = 3, **p < 0.01. (**D**) H&E staining showing similar histological structure of the testes from WT and *Xap5l* KO mice. Scale bars: 100 μm. (**E**) H&E staining showing the morphology of sperm in WT and *Xap5l* KO mice. Scale bars: 10 μm. (**F**) Significantly increased abnormal sperm in *Xap5l* KO mice. *n* = 6, ***p < 0.001. (**G**) Electron microscopic analysis displaying ultrastructural defects in *Xap5l* KO sperm flagella, including cytoplasmic vacuoles enveloping mitochondrial sheath (as shown by red arrowheads), a complete lack of mitochondrial sheath (as shown by red dotted circle), two or more cross-sections of the same sperm flagellum enclosed within one cell membrane (as shown by black arrowheads), partially formed outer dense fibers, and the axonemal microtubules (as indicated by asterisks). Scale bars: 2 μm. Error bars depict means ± SEM. All p-values were calculated using an unpaired, two-tailed Student's *t*-test.

The online version of this article includes the following video(s) for figure 2:

**Figure 2—video 1.** Representative video of living sperm cells from WT mouse.

*Figure 2 continued on next page*

*Figure 2 continued*

https://elifesciences.org/articles/94754/figures#fig2video1

**Figure 2—video 2.** Representative video of living sperm cells from *Xap5l* KO mouse.

https://elifesciences.org/articles/94754/figures#fig2video2

in *Xap5* cKO males was arrested in meiotic prophase, and Xap5 is indispensable for meiotic progression during spermatogenesis.

## Xap5 and Xap5l function as antagonistic regulators of ciliary transcriptional regulatory networks via regulating transcription factors

To gain insight into the regulatory mechanisms of Xap5l during sperm flagellar assembly, we employed RNA-seq analysis to examine mature sperm collected from the cauda epididymis of WT and *Xap5l* KO adult mice. Differential expression analysis identified 2093 upregulated and 267 downregulated genes in *Xap5l* KO sperm (*Figure 4—figure supplement 1A*). Gene ontology analysis revealed that the upregulated genes were associated with cilia assembly or function (*Figure 4A*, *Figure 4—figure supplement 1B, C*). When crossed with the previously compiled ciliogenesis-related gene list (*Nemajerova et al., 2016*), the upregulated genes were observed encompassing diverse structural and functional components of the ciliogenic program (*Supplementary file 1* and *Figure 4—figure supplement 1D*). Notably, several key transcription factor regulators of ciliogenesis were among the upregulated genes, including Foxj1 and Rfx families (*Figure 4B, C*, *Figure 4—figure supplement 1E*). Moreover, several key regulators of spermatogenesis, particularly spermiogenesis, were identified, including Rfx2 (*Kistler et al., 2015*; *Wu et al., 2016*), Sox families (*Schartl et al., 2018*; *Zhang et al., 2018*), Taf7l (*Zhou et al., 2013*), and Tbpl1 (*Martianov et al., 2001*; *Figure 4B*, *Figure 4—figure supplement 1E*). Many core ciliary genes were also upregulated, such as *Cfap206* (*Beckers et al., 2020*) and *Ift81* (*Qu et al., 2020*) which are critical for male fertility (*Figure 4—figure supplement 1E*), consistent with the tight coregulation between spermiogenesis and cilia-related genes during spermatogenesis. These findings suggest that the genes identified in our study provide an excellent resource for candidates with novel ciliary or spermatogenesis-related functions, and the newly identified testes-specific protein Tulp2 (*Oyama et al., 2022*; *Zheng et al., 2021*) was selected for validation. We observed that *Tulp2* KO male mice were sterile due to malformation of sperm flagella (*Figure 4—figure supplement 2*). Overall, our results clearly indicate that Xap5l acts as a central transcriptional repressor of ciliogenesis during mouse spermatogenesis.

Furthermore, we performed RNA-seq analysis using P16 testes samples from WT and *Xap5* cKO mice to explore the underlying mechanism of Xap5-mediated germ cell loss. Differential expression analysis identified 554 upregulated and 1587 downregulated genes in *Xap5* cKO sperm (*Figure 4—figure supplement 3A*). We observed that the downregulated genes in *Xap5* cKO males were involved in cilium assembly or functions (*Figure 4D*, *Figure 4—figure supplement 3B–D*). Strikingly, many of these downregulated genes were upregulated in *Xap5l* KO sperm, including the key ciliogenesis regulators Foxj1 and Rfx families (*Figure 4E, F*, *Figure 4—figure supplement 3E*, and *Supplementary file 1*). In addition, meiosis initiation marker Stra8 (*Anderson et al., 2008*; *Ferder et al., 2019*), meiosis-specific genes Spo11 (*Romanienko and Camerini-Otero, 2000*) and Dmc1 (*Bishop et al., 1992*), key transcription regulators of spermiogenesis Rfx2 (*Kistler et al., 2015*; *Wu et al., 2016*), Sox30 (*Zhang et al., 2018*), and Crem (*Blendy et al., 1996*; *Nantel et al., 1996*) were also downregulated in *Xap5* cKO mice (*Figure 4E*, *Figure 4—figure supplement 3F*), consistent with the aforementioned result that loss of Xap5 induced spermatogenesis arrest in meiotic stage.

To identify direct targets of Xap5/Xap5l, we conducted CUT&Tag analyses on P16 mouse testicular germ cells for Xap5, and on P56 mouse testicular germ cells for Xap5l. Our results revealed 23,143 peaks enriched for Xap5, compared to only 2290 peaks enriched for Xap5l (*Supplementary file 1*), and the limited number of enriched peaks for Xap5l suggests that the antibody used for Xap5l in the CUT&Tag experiment might have suboptimal performance. A substantial proportion of these peaks were located within promoter regions (*Figure 4—figure supplement 4A, B*), indicating that Xap5/Xap5l primarily function to regulate the expression of protein-coding genes. De novo motif analyses showed that the binding motifs for Xap5 and Xap5l shared a conserved sequence (CCCCGCCC/

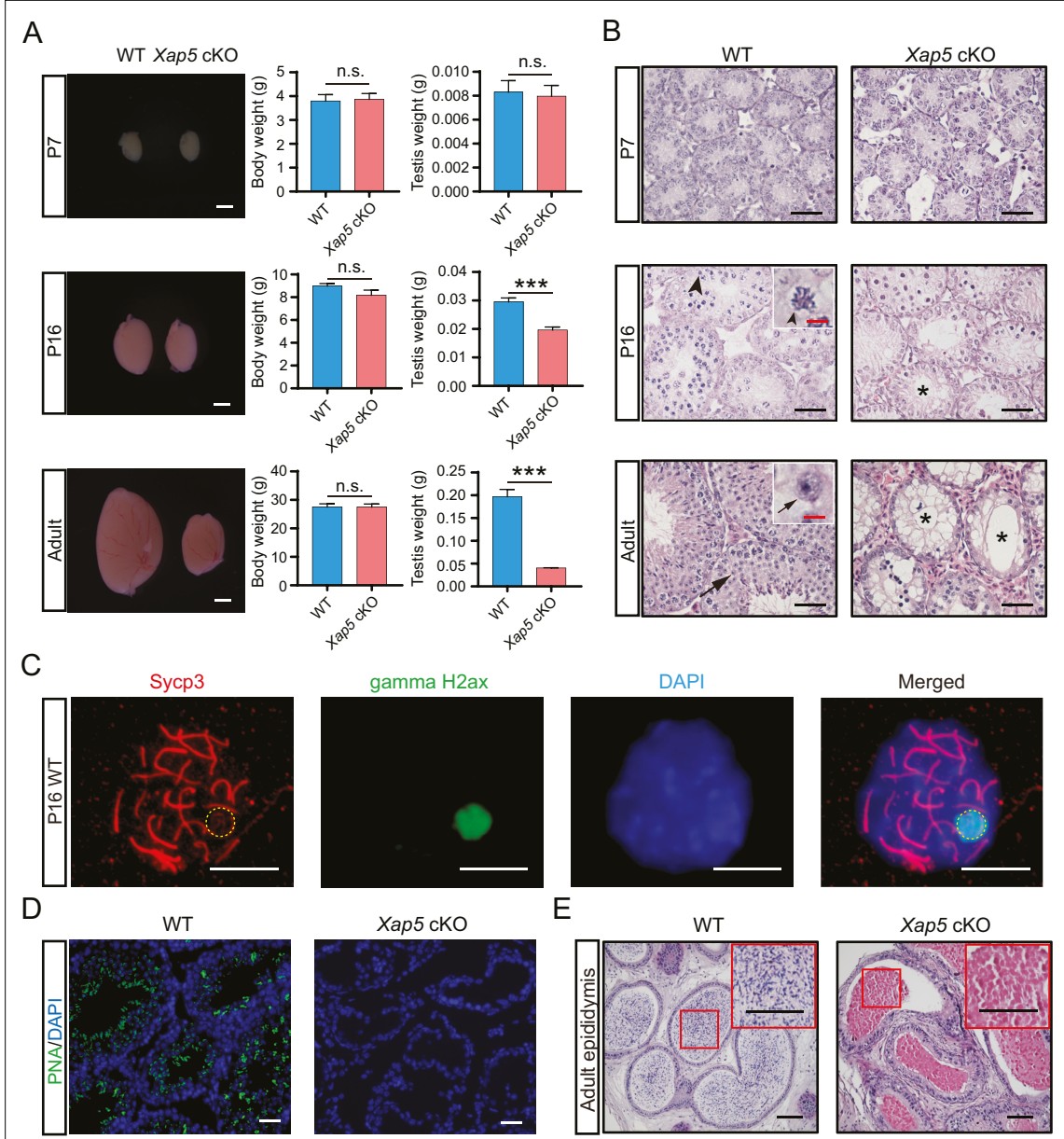

**Figure 3.** Ablation of Xap5 in germ cells results in arrested spermatogenesis and absence of sperm. (**A**) Testes sizes, body weight, and testis weight from WT and *Xap5* cKO mice at various ages were evaluated. Similar body weight was observed between WT and *Xap5* cKO mice. No significant difference in size and weight was detected between WT and *Xap5* cKO testes until P16. Scale bars: 1 mm. *n* = 5, ***p < 0.001, n.s. stands for not significant. (**B**) Histology of testes from WT and *Xap5* cKO mice at various ages was evaluated by H&E staining. No pachytene spermatocytes were found, and many vacuolated tubules were observed in *Xap5* cKO testis since P16. Black arrowheads indicate pachytene spermatocytes; black arrows indicate round spermatids. Vacuolated seminiferous tubules are indicated by asterisks. Scale bars: 20 μm (black), 2.5 μm (red). (**C**) Chromosome spreads staining showing pachytene spermatocytes from P16 WT male mouse. Yellow dotted circle indicates the XY body. Scale bars: 10 μm. (**D**) Immunofluorescence staining of PNA in adult testes indicating the absence of spermatids in *Xap5* cKO mice. Scale bars: 20 μm. (**E**) H&E staining of adult epididymis displaying absence of sperm in *Xap5* cKO mice. Scale bars: 50 μm. Error bars depict means ± SEM. All p-values were calculated using an unpaired, two-tailed Student's *t*-test.

GGGCGGGG) (*Figure 4—figure supplement 4C*), suggesting that Xap5 may regulate a subset of the same downstream genes as Xap5l.

To further elucidate the target genes of Xap5 and Xap5l, we integrated our CUT&Tag data with RNA-seq datasets (*Supplementary file 1*). This analysis revealed that among the genes downregulated in *Xap5* cKO germ cells, 891 were bound by Xap5 (*Figure 4—figure supplement 4D*). Similarly, 75 genes upregulated in *Xap5l* KO sperm were occupied by Xap5l (*Figure 4—figure supplement*

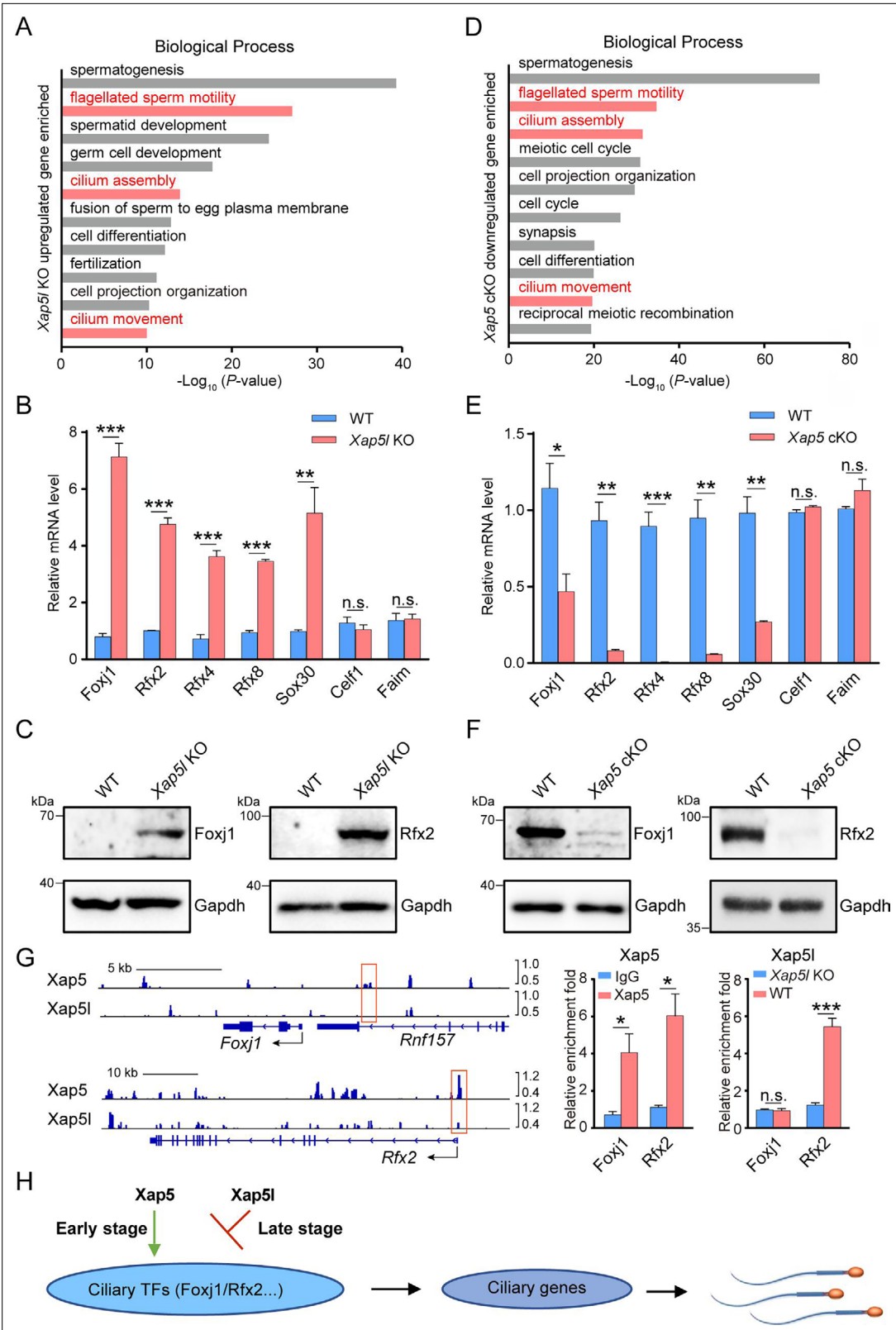

**Figure 4.** Aberrant ciliogenic and spermatogenic gene expression in *Xap5l* KO and *Xap5* cKO male germ cells. (**A**) Gene ontology analysis showing top 10 biological process terms significantly enriched among upregulated genes in *Xap5l* KO sperm. (**B**) Real-time PCR validation of genes involved in ciliogenesis and spermatogenesis. *n* = 3, **p < 0.01, ***p < 0.001, n.s. stands for not significant. (**C**) Detection of Foxj1 and Rfx2 in mature sperm of WT and *Xap5l* KO mice by western blot analysis. Gapdh was used as a control. (**D**) Gene ontology analysis showing top 10 biological process

*Figure 4 continued on next page*

*Figure 4 continued*

terms significantly enriched among downregulated genes in P16 *Xap5* cKO testes. (**E**) Real-time PCR validation of genes involved in ciliogenesis and spermatogenesis. $n = 3$, *$p < 0.05$, **$p < 0.01$, ***$p < 0.001$, n.s. stands for not significant. (**F**) Detection of Foxj1 and Rfx2 in P16 testes of WT and *Xap5* cKO mice by western blot analysis. Gapdh was used as a control. (**G**) Xap5/Xap5l occupancies at *Foxj1* and *Rfx2* promoters assessed by CUT&Tag-seq (left) and CUT&Tag real-time PCR (middle and right). Red boxes indicate the binding regions of Xap5 and Xap5l. $n = 3$, *$p < 0.05$, ***$p < 0.001$, n.s. stands for not significant. (**H**) A model on the involvement of Xap5 and Xap5l in the regulation of ciliogenesis during spermatogenesis. Error bars depict means ± SEM. All p-values were calculated using an unpaired, two-tailed Student's *t*-test.

The online version of this article includes the following source data and figure supplement(s) for figure 4:

**Source data 1.** Original files for western blot analysis displayed in *Figure 4C and F*.

**Source data 2.** File containing the original western blots for Figure *4C and F*, indicating the relevant bands and corresponding samples.

**Figure supplement 1.** Xap5l-mediated regulation of ciliary gene expression during mouse spermatogenesis.

**Figure supplement 2.** *Tulp2* KO male mice were infertile due to malformation of sperm.

**Figure supplement 2—source data 1.** Original files for western blot analysis displayed in *Figure 4—figure supplement 2A*.

**Figure supplement 2—source data 2.** File containing the original SDS-PAGE gel and western blot images for *Figure 4—figure supplement 2A*, indicating the relevant bands and corresponding samples.

**Figure supplement 3.** Xap5-mediated regulation of ciliary gene expression during mouse spermatogenesis.

**Figure supplement 4.** Identification of Xap5/Xap5l-occupancy sites.

**Figure supplement 5.** t-SNE plots displaying expression patterns of *Foxj1* and *Rfx* families in individual cell types of mouse testes.

*4E*). GO analysis indicated that these genes are associated with cilium assembly and related functions (*Figure 4—figure supplement 4D–G*). Notably, both *Foxj1* and *Rfx2* are regulated and occupied by Xap5, with the conserved sequence present within the Xap5-bound regions (*Figure 4G*). Furthermore, real-time PCR validation of the CUT&Tag experiments confirmed that *Rfx2* is also occupied by Xap5l, despite the initial CUT&Tag data not showing enriched peaks for the Rfx2 gene (*Figure 4G*). Unfortunately, due to the suboptimal performance of the Xap5l antibody, we did not detect the occupancy of Xap5l on the Foxj1 promoter. Nevertheless, these findings suggest that Xap5 and Xap5l regulate ciliary gene expression primarily through direct binding to the conserved motif present in these genes.

Considering the evolutionary conservation of Xap5/Xap5l protein across different species and the expression patterns of Xap5, Xap5l, Foxj1, and Rfx factors during spermatogenesis (*Figure 1C*, *Figure 4—figure supplement 5*), these results suggest that Xap5 and Xap5l have antagonistic effects, and they may function upstream of the transcription factor families, including Foxj1 and Rfx factors, to coordinate the ciliogenesis during spermatogenesis (*Figure 4H*).

## Discussion

Cilia and flagella are ancient organelles present in the last eukaryotic common ancestor (*Mitchell, 2017*). Cilia assembly and maintenance are under strict transcriptional regulation in both unicellular and multicellular organisms (*Choksi et al., 2014*; *Collins et al., 2021*; *Lewis and Stracker, 2021*; *Thomas et al., 2010*). Despite cilia and flagella having an ancient origin, the evolutionary history of ciliary gene regulation has remained an unsolved problem. The main reason is that the master ciliary transcription factors found in multicellular organisms, including Foxj1 and Rfxs, are absent from unicellular organisms (*Choksi et al., 2014*; *Collins et al., 2021*; *Lewis and Stracker, 2021*; *Thomas et al., 2010*). Previously, we showed that an ancient transcription factor, Xap5, was required for flagella assembly in the unicellular green algae *Chlamydomonas* (*Li et al., 2018*). Xap5 proteins are evolutionarily conserved across diverse organisms (*Li et al., 2018*; *Martin-Tryon and Harmer, 2008*), which offers the possibility to investigate the conservation of the role of Xap5 in multicellular organisms. In the present study, we report that Xap5 positively regulates transcriptional network of ciliary genes by activating the key regulators including Foxj1 and Rfxs during spermatogenesis.

Cilia are dynamic organelles, and the expression profile of ciliary genes is dynamic during the cell cycle and under certain physiological conditions (*Eggenschwiler and Anderson, 2007*; *Kasahara and Inagaki, 2021*; *Santos and Reiter, 2008*). Cells have developed regulatory mechanisms to generate functional cilia in a temporal and spatial manner (*Choksi et al., 2014*; *Collins et al., 2021*; *Lewis and Stracker, 2021*; *Thomas et al., 2010*). However, how ciliary transcription factors determine temporal and spatial ciliary transcriptional programs is poorly understood. Sperm flagellar assembly is likewise

tightly regulated during a highly complex temporal process named spermatogenesis (*Holstein et al., 2003*; *Thomas et al., 2010*). Several key transcriptional regulators of spermatogenesis, such as Foxj1 and Rfx2, have been identified (*Chen et al., 1998*; *Kistler et al., 2015*; *Wu et al., 2016*). The flagellar formation of Foxj1-null sperm is severely impaired, and the Foxj1 target genes are required for sperm motility (*Beckers et al., 2020*; *Chen et al., 1998*; *Weidemann et al., 2016*). *Rfx2* KO mice are completely sterile and Rfx2 regulates the expression of ciliary genes during spermiogenesis (*Kistler et al., 2015*; *Wu et al., 2016*). Ciliary transcription factors (Foxj1 and Rfx2) are dynamically regulated during spermatogenesis (*Jung et al., 2019*). Intriguingly, our results show that Xap5 and Xap5l are two conserved pairs of antagonistic transcription factors that regulate these key transcriptional regulators required for spermatogenesis.

By using western blot analysis, we found that Xap5 was ubiquitously expressed in all tissues examined, whereas Xap5l expression was restricted to the testes (*Figure 1A*). Interestingly, previous research found that Xap5l was widely expressed in normal tissue via using TCGA expression data (*Thompson et al., 2021*). Moreover, loss of Xap5/Xap5l perturbs transcriptional programs in cancer cells. At present, as shown by the recent data, we cannot rule out the possibility that Xap5l is dynamically expressed in other tissues during development. Although Xap5 and Xap5l are highly expressed in testes and critical for spermatogenesis, they are not present in mature sperm (*Figure 1—figure supplement 1*). Thus, Xap5/Xap5l are not needed for sperm maintenance. Given the dynamic expression of Xap5/Xap5l during spermatogenesis, it will be interesting to identify the key signaling factors that regulate the timely and spatial expression of Xap5 and Xap5l.

A role for Xap5 in human brain development is suggested by the association of X-linked intellectual disability (XLID) with rare *Xap5* missense variants (*Lee et al., 2020*). However, how defects in Xap5 lead to XLID syndrome is unknown. Although there is as yet no evidence that Xap5 regulates ciliogenesis in any system besides *Chlamydomonas* and mice, the function in cilia could help clarify the unexplained role of *Xap5* mutations in causing XLID. The possibility that Xap5 could be a ciliary transcription factor in humans would add XLID to the growing list of the second-order ciliopathies.

## Materials and methods

### Mice

C57BL/6J strains (from Vital River Laboratories, Beijing, China) and *Stra8-GFPCre* knockin mice (from Cyagen Biosciences) were used. Floxed-*Xap5* mice, *Xap5l* and *Tulp2* knockout mice were generated on the C57BL/6J background using the CRISPR/Cas9 system. The sgRNA sequences used to target *Xap5*, *Xap5l*, and *Tulp2* were as follows: *Xap5l*, sgRNA1-GGC TAC CAG AAA CAG GGA CT, sgRNA2-GGG AGT AAG GTC CCC AAA CT; *Xap5*, sgRNA3-CTA CAG GGC ACT TAT TAA TA, sgRNA4-ATA GTA ATT CCC CCG TGC TT; *Tulp2*, sgRNA5-TGA CTA ATT AGG CCC GAG AG, sgRNA6-GGT TCT TAG AGA GTC AAC GT. Experimental protocols were approved by the ethics committees of Jianghan University (number: JHDXLL2024-039). Mice were maintained in a pathogen-free environment with a room temperature of 23 ± 2°C under a humidity level of 30–70%. The light was maintained on a 12-hr day/night cycle.

### Genome PCR

Mouse genomic DNA was isolated from the tail tip using One Step Mouse Genotyping Kit (Vazyme, PD101), and subjected to PCR with 2 × Taq Plus Master Mix (Vazyme, P212) following the manufacturer's instructions. The PCR products were separated by 2% agarose gel electrophoresis. The primers used are listed in *Supplementary file 2*.

### RNA isolation and real-time PCR

Total RNA was isolated using the TRIzol reagent (Invitrogen, 15596018) and then converted to cDNA with HiScript II 1st Strand cDNA Synthesis Kit (Vazyme, R212) following the manufacturer's instructions. Real-time PCR was conducted using ChamQ SYBR qPCR Master Mix (Vazyme, Q311) on an ABI StepOnePlus real-time PCR system (Applied Biosystems). Expression values were normalized using the ΔΔCt method with *Gapdh* as the internal control. The primer sequences are indicated in *Supplementary file 2*.

## Western blot analysis

Cell lysates from mouse tissues were prepared in RIPA buffer (Beyotime, P0013B) containing Protease Inhibitor Cocktail (Roche, 11873580001) and 1 mM Phenylmethanesulfonyl fluoride (PMSF, Sigma, 78830). Subcellular lysates of testicular nuclear and cytoplasmic fractions were prepared using Nuclear and Cytoplasmic Protein Extraction Kit (Beyotime, P0027) following the manufacturer's instructions. The protein lysates were subjected to standard immunoblotting analysis. Antibodies used were as follows: anti-β-actin (ABclonal, AC026), anti-Xap5 (Prospertech, HU-412003), anti-Xap5 (Sigma, HPA003585), anti-Xap5l (Prospertech, Hu-412001), anti-Lmnb (Proteintech, 66095–1-Ig), anti-Gapdh (ABclonal, AC002), anti-Foxj1 (D360355, Sangon Biotech), anti-Rfx2 (K110716P, Solarbio), and anti-Tulp2 (Prospertech, Hu-412002).

## Immunofluorescence

Mouse testis were fixed in 4% paraformaldehyde in PBS at 4°C overnight, dehydrated in gradient ethanol, embedded in paraffin, and sectioned into 5 μm slices. After deparaffinization and rehydration, testis sections were boiled in 1 mM EDTA, pH 8.0 for 15 min using a microwave oven for antigen retrieval, followed by cooling to room temperature. The primary antibodies, including anti-Xap5 (Prospertech, HU-412003, 1:250 dilution), anti-Xap5 (Sigma, HPA003585, 1:250 dilution), anti-Xap5l (Prospertech, Hu-412001, 1:250 dilution), anti-lectin PNA, Alexa Fluor 488 Conjugate (Thermo Fisher, L21409, 1:250 dilution), and anti-Tulp2 (Prospertech, Hu-412002, 1:250 dilution), were then incubated at 4°C overnight. Subsequently, the sections were incubated with the secondary antibody donkey anti-rabbit Alexa Fluor Plus 488 (Invitrogen, A32790, 1:500 dilution) for 1 hr and DAPI for 5 min at room temperature. Images were collected using a Zeiss Axio Vert A1 microscope.

## Histological analysis

Testes and epididymides were fixed in Bouin's solution (sigma, HT10132) overnight at 4°C, embedded in paraffin and sectioned into 5 μm slices. The sections were then subjected to H&E staining using standard procedures and observed under a Zeiss Axio Vert A1 microscope.

## Assessment of sperm counts, motility, and morphology

Cauda epididymides were collected from male mice and dissected in 37°C pre-warmed human tubal fluid culture medium (EasyCheck, M1130), followed by incubation at 37°C for 15 min to release the sperm. To quantify sperm abundance, the sperm suspension was fixed in 1% paraformaldehyde for 30 min. Sperm were then counted under a Zeiss ×10 bright-field microscope. To assess sperm motility, the sperm suspension was observed and recorded under a Leica total internal reflection fluorescence microscope. Motility was defined as any movement of the sperm flagellum during a 10-s observation cycle. After immobilization with 1% paraformaldehyde, sperm morphology was observed by H&E staining using standard procedures. Images were captured using a Zeiss Axio Vert A1 microscope equipped with a ×100 oil immersion objective. Sperm without staining were also used to assess sperm morphology. For ultrastructural observation, sperm samples were fixed in 3% glutaraldehyde in 0.1 M sodium phosphate buffer (pH 7.4), and post-fixed with 1% (wt/vol) $OsO_4$. After dehydration, the samples were placed in propylene oxide and embedded in a mixture of Epon 812 and Araldite. Ultrathin sections obtained by a Leica EM UC7 ultramicrotome were stained with uranyl acetate and lead citrate, then analyzed using a HT7700 TEM (Hitachi).

## Fertility test

Each of the adult male mice was mated with two females for 2 months. All the mice used were aged 6–8 weeks. Vaginal plugs were checked every morning, and the number of newborn pups was counted.

## Chromosome spreads and immunofluorescence

Testicular chromosome spreads were prepared and immunolabeled as described (*Reinholdt et al., 2004*). The primary antibodies used were as follows: anti-γ H2ax (Millipore, 05-636, 1:300 dilution) and anti-Sycp3 (Proteintech, 23024-1-AP, 1:300 dilution). Secondary antibodies used were Alexa Dye (AlexaFluor Plus 488/594) conjugates (Thermo Fisher) at 1:500 dilutions. Images were collected with a Zeiss Axio Vert A1 microscope.

## RNA-seq analysis

The mature sperm samples from P56 mice and testis samples from P16 mice were washed three time in 1× PBS after harvest. Total RNA was extracted from the samples using the TRIzol reagent (Invitrogen) and purified with the NEBNext Poly(A) mRNA Magnetic Isolation Module (NEB, E7490). The quality of the RNA samples was examined using the Agilent 2100 Bioanalyzer system (Agilent Technologies). Qualified RNA samples were subjected to sequencing library construction using the NEBNext Ultra II RNA Library Prep Kit for Illumina (NEB, E7775). Paired-end (2 × 150 bp) sequencing was carried out on the Illumina NovaSeq6000 platform. The sequencing reads were mapped to the *Mus musculus* reference genome (GRCm38/mm10) using HISAT2 v2.1.0 (*Kim et al., 2015*), and StringTie v1.3.3b (*Pertea et al., 2015*) was applied to assemble the mapped reads. Fragments Per Kilobase of transcript per Million fragments mapped (FPKM) was used to measure the expression level of a gene. Differential expression analysis was processed by DESeq2 v1.12.4 (*Subramanian et al., 2005*), and the significant differentially expressed genes were identified with a fold change >2. Gene ontology analysis was performed using the DAVID database.

## CUT&Tag analysis

The testicular germ cells were isolated using a two-step enzymatic digestion process as previously described (*Liu et al., 2017*). Briefly, after removing the tunica albuginea, the testes were incubated in 1 mg/ml type IV collagenase (Sigma, C5138) at 37°C for 15 min with gentle shaking every 5 min. The resulting cell suspension was centrifuged at 1000 rpm for 3 min, and the pellet was further digested with 0.25% Trypsin-EDTA (Gibco, 25200072) at 37°C for 10 min. The digestion was terminated by adding fetal bovine serum (FBS), and the cell suspension was centrifuged at 1000 rpm for 3 min. The cell pellet was resuspended in DMEM medium containing 10% FBS and cultured at 37°C in a 5% $CO_2$ atmosphere for 3 hr. Floating and weakly adherent cells were collected as germ cells. CUT&Tag and CUT&Tag real-time PCR were performed according to the manufacturer's protocol using the Hyperactive Universal CUT&Tag Assay Kit for Illumina Pro (Vazyme, TD904). Approximately 100,000 germ cells were used per sample. For real-time PCR, IgG sera were used as a negative control for the Xap5 CUT&Tag, while lysates from *Xap5l* KO testis were incubated with the Xap5l antibody as a negative control for the Xap5l CUT&Tag. DNA spike in was used as the internal control. The primer sequences are indicated in *Supplementary file 2*.

For analysis, quality control and read trimming were performed using FastQC (v0.11.9) with default parameters and Trimmomatic (v0.39) with the following parameters: ILLUMINACLIP:NexteraPE-PE.fa: 2:30:10:8:true LEADING:3 TRAILING:3 SLIDINGWINDOW:4:15 MINLEN:8. Clean reads were aligned with bowtie2 (v2.4.4) (*Langmead and Salzberg, 2012*) against the *M. musculus* (mm39) genome with the parameters -X 1000 `--very-sensitive`. PCR duplicates were removed using picard's MarkDuplicates with default parameters. Peak calling was conducted using using MACS2 (v2.1.4) with the parameters -f BAMPE -B `--SPMR --keep-dup` all. Peaks were annotated to the nearest feature using ChIPseeker (v1.32.0) (*Yu et al., 2015*). Motif finding and annotation were performed using Homer *findMotifsGenome.pl* and *annotatePeaks.pl* functions, respectively (*Heinz et al., 2010*). Track screen shots were produced in IGV (version 2.18.1) (*Robinson et al., 2023*).

## Statistical analysis

Statistical analyses were conducted using GraphPad Prism 8.0.2 software. All data were presented as mean ± SEM. Values of $p < 0.05$ were considered statistically significant. Statistical significance between two groups was calculated using an unpaired, parametric, two-sided Student's *t*-test.

## Materials availability

The authors declare that all the materials supporting the findings of this study are available within the article and its supplementary files or from the corresponding author upon reasonable request.

## Acknowledgements

This work was supported by the National Key R&D Program of China (2020YFA0907400), the National Natural Science Foundation of China (32170702 and 82000828), and the Major Special Funding Program for First-class Discipline Construction of Jianghan University (2023XKZ021). We thank Yuan

He (Research Center for Medicine and Structural Biology, Wuhan University, China) for the assistance during the experimental design of the transmission electron microscope analysis.

## Additional information

### Funding

| Funder | Grant reference number | Author |
|---|---|---|
| National Key Research and Development Program of China | 2020YFA0907400 | Zhangfeng Hu |
| National Natural Science Foundation of China | 32170702 | Zhangfeng Hu |
| National Natural Science Foundation of China | 82000828 | Haochen Jiang |
| Major Special Funding Program for First-class Discipline Construction of Jianghan University | 2023XKZ021 | Zhangfeng Hu |

The funders had no role in study design, data collection and interpretation, or the decision to submit the work for publication.

### Author contributions

Weihua Wang, Conceptualization, Data curation, Supervision, Investigation, Methodology, Writing – original draft, Project administration, Writing – review and editing; Junqiao Xing, Xiqi Zhang, Hongni Liu, Data curation, Investigation, Methodology; Xingyu Liu, Cheng Xu, Investigation, Methodology; Haochen Jiang, Funding acquisition, Investigation; Xue Zhao, Investigation; Zhangfeng Hu, Conceptualization, Data curation, Supervision, Funding acquisition, Writing – original draft, Project administration, Writing – review and editing

### Author ORCIDs

Weihua Wang https://orcid.org/0009-0007-3965-7375
Haochen Jiang https://orcid.org/0009-0003-4254-6765
Zhangfeng Hu https://orcid.org/0000-0002-4841-8073

### Ethics

All experiments were performed in accordance with standard ethical guidelines and were approved by the ethics committees of Jianghan University (number: JHDXLL2024-039).

Reviewer #1 (Public review): https://doi.org/10.7554/eLife.94754.3.sa1
Reviewer #2 (Public review): https://doi.org/10.7554/eLife.94754.3.sa2
Author response https://doi.org/10.7554/eLife.94754.3.sa3

## Additional files

### Supplementary files

Supplementary file 1. The lists of differential expression genes, ciliogenesis-related genes, and enriched peaks for Xap5/Xap5l.

Supplementary file 2. The lists of primers used in the genome PCR and real-time PCR.

MDAR checklist

### Data availability

The RNA-seq and CUT&Tag data generated in this study have been deposited in the Gene Expression Omnibus (GEO) database and are publicly available under accession numbers GSE236388 for RNA-seq and GSE279586 for CUT&Tag, respectively.

The following datasets were generated:

| Author(s) | Year | Dataset title | Dataset URL | Database and Identifier |
|---|---|---|---|---|
| Wang W, Xing J, Zhang X, Hu Z | 2023 | Control of ciliary transcriptional programs during spermatogenesis by antagonistic transcription factors | https://www.ncbi.nlm.nih.gov/geo/query/acc.cgi?acc=GSE236388 | NCBI Gene Expression Omnibus, GSE236388 |
| Wang W, Xing J, Zhang X | 2024 | Identification of chromatin states in mouse testicular germ cells using CUT&Tag | https://www.ncbi.nlm.nih.gov/geo/query/acc.cgi?acc=GSE279586 | NCBI Gene Expression Omnibus, GSE279586 |

The following previously published dataset was used:

| Author(s) | Year | Dataset title | Dataset URL | Database and Identifier |
|---|---|---|---|---|
| Jung M, Wells DJ, Rusch J, Ahmad S, Marchini J, Myers S, Conrad DF | 2019 | Phenotyping spermatogenic defects by single-cell expression profiling | https://www.ncbi.nlm.nih.gov/geo/query/acc.cgi?acc=GSE113293 | NCBI Gene Expression Omnibus, GSE113293 |

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
