## [Editor Report · eLife Assessment]

This **valuable** study presents data suggesting the critical roles of two ancient proteins, XAP5 and XAP5L, in regulating the transcriptional program of ciliogenesis during mouse spermatogenesis. The supporting data are **solid**, and this work will be of interest to biomedical researchers studying ciliogenesis and reproduction.

---

## [Referee Report · Reviewer #1 (Public review)]

Summary:

Wang et al. generate XAP5 and XAP5L knockout mice and find that they are male infertile due to spermatogonial/meiotic arrest and reduced sperm motility, respectively. CUT & Tag data were added in this revision in order to support that XAP5 and XAP5L are antagonistic transcription factors of cilliogenesis.

Strengths:

Knockout mouse models provided strong evidence to indicate that XAP5 and XAP5L are critical for spermatogenesis. RNA-seq and CUT & Tag are valuable sources to further explore their molecular mechanisms.

Weaknesses:

The authors claim that XAP5 and XAP5L transcriptionally regulate sperm flagella development; however, expression, physiological role, and molecular evidence do not well support this concept. This reviewer still thinks the physiological roles of XAP5 and XAP5l are different. (i) XAP5 is expressed at spermatogonia within testes while XAP5l is localized at round/elongating spermatids (their expressions are different). (ii) Spermatogenesis was arrested at spermatogonia/early spermatocyte stage in Xap5-KO mice while sperm abnormalities were observed in Xap5l-KO mice (their roles are different). This reviewer still can't get the authors' viewpoint that XAP5 and XAP5l are 'antagonistic relationship' to regulate sperm flagella development. RNA-seq and CUT & Tag data are valuable source; however, this reviewer suggests exploring how XAP5 regulates spermatogonia differentiation and how XAP5l regulates sperm flagella motility.

---

## [Referee Report · Reviewer #2 (Public review)]

In this study, Wang et al., report the significance of XAP5L and XAP5 in spermatogenesis which are involved in transcriptional regulation of the ciliary gene in testes. In a previous study, the authors demonstrated that XAP5 is a transcription factor required for flagellar assembly in Chlamydomonas. Continuing from their previous study, the authors examined conserved role of the XAP5 and XAP5L, which are the orthologue pair in mammals.

XAP5 and XAP5L express ubiquitously and testis specifically, respectively, and their absence in testes causes male infertility with defective spermatogenesis. Interestingly, XAP5 deficiency arrest germ cell development at pachytene stage, whereas XAP5L absence causes impaired flagellar formation. RNA-seq analyses demonstrated that XAP5 deficiency suppresses ciliary gene expression including Foxj1 and Rfx family genes in early testis. By contrast, XAP5L deficiency abnormally remains Foxj1 and Rfx genes in mature sperm. From the results, the authors conclude that XAP5 and XAP5L are the antagonistic transcription factor to function at the upstream of Foxj1 and Rfx family genes.

The current version of the manuscript well represents this reviewer's initial concerns and supports author's claim. Key transcription factors for ciliogenesis, Foxj1 and Rfx2, are direct downstream targets for XAP5 and XAP5L and their common motifs well explain their antagonistic function in sperm flagellar development. All the results well demonstrate that ancient transcription factors, XAP5 and XAP5L, are upstream transcription factors to modulate flagellar development in male mammalian germ line.

---

## [Author Response]

The following is the authors’ response to the original reviews.

**Public Reviews:**

**Reviewer #1 (Public Review):**
Summary:Wang et al. generate XAP5 and XAP5L knockout mice and find that they are male infertile due to meiotic arrest and reduced sperm motility, respectively. RNA-Seq was subsequently performed and the authors concluded that XAP5 and XAP5L are antagonistic transcription factors of cilliogenesis (in XAP5-KO P16 testis: 554 genes were unregulated and 1587 genes were downregulated; in XAP5L-KO sperm: 2093 genes were unregulated and 267 genes were downregulated).

We are grateful for the comprehensive summary.

Strengths:Knockout mouse models provided strong evidence to indicate that XAP5 and XAP5L are critical for spermatogenesis and male fertility.

Thank you for your positive comment.

Weaknesses:The key conclusions are not supported by evidence. First, the authors claim that XAP5 and XAP5L transcriptionally regulate sperm flagella development; however, detailed molecular experiments related to transcription regulation are lacking. How do XAP5 and XAP5L regulate their targets? Only RNA-Seq is not enough. Second, the authors declare that XAP5 and XAP5L are antagonistic transcription factors; however, how do XAP5 and XAP5L regulate sperm flagella development antagonistically? Only RNA-Seq is not enough. Third, I am concerned about whether XAP5 really regulates sperm flagella development. XAP5 is specifically expressed in spermatogonia and XAP5-cKO mice are in meiotic arrest, indicating that XAP5 regulates meiosis rather than sperm flagella development.

Thank you for the critical comments. To strengthen our conclusions, we have included XAP5/XAP5L CUT&Tag data in our revised manuscript. This highly sensitive method has allowed us to identify direct target genes of XAP5 and XAP5L (Table S1, Figure S6). Notably, our results demonstrate that both *FOXJ1* and *RFX2* are occupied by XAP5 (Figure 4G). Additionally, real-time PCR validation confirmed that RFX2 is also associated with XAP5L, even though enriched peaks for the RFX2 gene were not detected in the initial CUT&Tag data (Figure 4G). These findings indicate that XAP5 and XAP5L regulate the expression of FOXJ1 and RFX2 by directly binding to these genes. De novo motif analyses revealed that XAP5 and XAP5L shared a conserved binding sequence (CCCCGCCC/GGGCGGGG) (Figure S6C), and the bound regions of *FOXJ1* and *RFX2* contain this sequence. Further analysis shows that many XAP5L target genes are also targets of XAP5 (Figure S6G), despite the limited number of identified XAP5L target genes. This differential binding and regulation of shared target genes underscore the antagonistic relationship between XAP5 and XAP5L. Collectively, these findings provide additional support for the idea that XAP5 and XAP5L function as antagonistic transcription factors, acting upstream of transcription factor families, including FOXJ1 and RFX factors, to coordinate ciliogenesis during spermatogenesis.

While we agree that XAP5 primarily regulates meiosis during spermatogenesis, our data also indicate that many cilia-related genes, including key transcription regulators of spermiogenesis such as RFX2 and SOX30, are downregulated in XAP5-cKO mice and are bound by XAP5 (Figure 4, Figures S4 and S6). It is important to note that genes coding for flagella components are expressed sequentially and in a germ cell-specific manner during development. When we refer to "regulating sperm flagella development", we mean the spatiotemporal regulation. We have revised the manuscript to clarify this point.

**Reviewer #2 (Public Review):**
In this study, Wang et al., report the significance of XAP5L and XAP5 in spermatogenesis, involved in transcriptional regulation of the ciliary gene in testes. In previous studies, the authors demonstrate that XAP5 is a transcription factor required for flagellar assembly in Chlamydomonas. Continuing from their previous study, the authors examine the conserved role of the XAP5 and XAP5L, which are the orthologue pair in mammals.XAP5 and XAP5L express ubiquitously and testis specifically, respectively, and their absence in the testes causes male infertility with defective spermatogenesis. Interestingly, XAP5 deficiency arrests germ cell development at the pachytene stage, whereas XAP5L absence causes impaired flagellar formation. RNA-seq analyses demonstrated that XAP5 deficiency suppresses ciliary gene expression including Foxj1 and Rfx family genes in early testis. By contrast, XAP5L deficiency abnormally remains Foxj1 and Rfx genes in mature sperm. From the results, the authors conclude that XAP5 and XAP5L are the antagonistic transcription factors that function upstream of Foxj1 and Rfx family genes.This reviewer thinks the overall experiments are performed well and that the manuscript is clear. However, the current results do not directly support the authors' conclusion. For example, the transcriptional function of XAP5 and XAP5L requires more evidence. In addition, this reviewer wonders about the conserved XAP5 function of ciliary/flagellar gene transcription in mammals - the gene is ubiquitously expressed despite its functional importance in flagellar assembly in Chlamydomonas. Thus, this reviewer thinks authors are required to show more direct evidence to clearly support their conclusion with more descriptions of its role in ciliary/flagellar assembly.

Thank you for your thoughtful review of our work. We appreciate your positive feedback on the overall quality of the experiments and the clarity of the manuscript. In response to your concerns, we have included new experimental data and made revisions to the manuscript (lines 193-217) to better support our conclusions, particularly regarding the transcriptional function of XAP5 and XAP5L. Additionally, we have expanded on the role of XAP5 in ciliary and flagellar assembly to provide more direct evidence for its functional importance. Thank you for your insights.

**Recommendations for the authors:**

**Reviewer #1 (Recommendations For The Authors):**
The title (Control of ciliary transcriptional programs during spermatogenesis by antagonistic transcription factors) is not specific and does tend to exaggerate.

Thank you for the comment, and we appreciate the opportunity to clarify the appropriateness of the title. Our paper extensively investigates the transcriptional regulation of ciliary genes during spermatogenesis. It demonstrates that XAP5/XAP5L are key transcription factors involved in this process. The title reflects our primary focus on the transcriptional programs that govern ciliary gene expression. Moreover, our paper shows that XAP5 positively regulates the expression of ciliary genes, particularly during the early stages of spermatogenesis, while XAP5L negatively regulates these genes. This antagonistic relationship is a crucial aspect of the study and is effectively conveyed in the title. In addition, our revised paper provides detailed insights into how XAP5/XAP5L control ciliary gene expression during spermatogenesis.

Figure 4C: FOXJ1 and RFX2 are absent in sperm from WT mice. Are you sure? They are highly expressed in WT testes.

Thank you for your careful review. While FOXJ1 and RFX2 are indeed highly expressed in the testes of wild-type (WT) mice, our data show that they are not detectable in mature sperm. This observation is consistent with published single-cell RNA-seq data(Jung et al., 2019), which indicate that FOXJ1 and RFX2 are primarily expressed in spermatocytes but not in spermatids (Figure S7). This expression pattern aligns with that that of IFT-particle proteins, which are essential for the formation but not the maintenance of mammalian sperm flagella(San Agustin, Pazour, & Witman, 2015).

XAP5 is specifically expressed in spermatogonia and XAP5-cKO mice are in meiotic arrest, indicating that XAP5 regulates meiosis rather than sperm flagella development.

We appreciate your insightful comments. As mentioned above, we agree that XAP5 primarily regulates meiosis during spermatogenesis. When we mentioned "regulating sperm flagella development," we were referring to the spatiotemporal regulation of these processes. We have revised the manuscript to clarify this distinction. Thank you for your understanding.

The title of Figure 2 (XAP5L is required for normal sperm formation) is not accurate because the progress of spermatogenesis and sperm count is normal in XAP5L-KO mice (only sperm motility is reduced).

We apologize for any confusion caused by the previous figure. It did not accurately convey the changes in sperm count. In the revised Figure 2B, we clearly demonstrate that the sperm count in XAP5L-KO mice is indeed lower than that in WT mice. This revision aims to provide a more accurate representation of the effects of XAP5L deficiency on spermatogenesis. Thank you for bringing this to our attention.

**Reviewer #2 (Recommendations For The Authors):**
(1) Although XAP5 and XAP5L deficiency alters the transcription of Foxj1 and Rfx family genes, which are the essential transcription factors for the ciliogenesis, current data do not directly support that XAP5 and XAP5L are the upstream transcription factors. The authors need to show more direct evidence such as CHIP-Seq data.

Thank you for your valuable feedback! In this revised manuscript, we have included data identifying candidate direct targets of XAP5 and XAP5L using the highly sensitive CUT&Tag method (Kaya-Okur et al., 2019). Our results show that XAP5 occupies both FOXJ1 and RFX2 (Figure 4G). Furthermore, real-time PCR validation of the CUT&Tag experiments confirmed that RFX2 is also occupied by XAP5L (Figure 4G), despite the initial CUT&Tag data not revealing enriched peaks for the RFX2 gene (Table S1). Unfortunately, the limited number of enriched peaks identified for XAP5L (Table S1) suggests that the XAP5L antibody used in the CUT&Tag experiment might have suboptimal performance, which prevented us from detecting occupancy on the FOXJ1 promoter. Nevertheless, these additional data provide strong evidence that XAP5 and XAP5L function as upstream transcription factors for FOXJ1 and RFX family genes, supporting their essential roles in ciliogenesis.

(2) Shared transcripts that are altered by the absence of either XAP5 or XAP5L do not clearly support they are antagonistic transcription factors.

Thank you for your insightful comment. In our revised manuscript, we performed CUT&Tag analysis to identify target genes of XAP5 and XAP5L. Motif enrichment analysis revealed conserved binding sequences for both factors (Figures S6C), indicating a subset of shared downstream genes between XAP5 and XAP5L. Among the downregulated genes in XAP5 cKO germ cells, 891 genes were bound by XAP5 (Figure S6D). Although the number of enriched peaks identified for XAP5L was limited, 75 of the upregulated genes in XAP5L KO sperm were bound by XAP5L (Figure S6E). Importantly, of these 75 XAP5L target genes, approximately 30% (22 genes) were also identified as targets of XAP5 (Figure S6G), further support the idea that XAP5 and XAP5L function as antagonistic transcription factors.

(3) XAP5 seems to be an ancient transcription factor for cilia and flagellar assembly. However, XAP5 expresses ubiquitously in mice. How can this discrepancy be explained? Is it also required for primary cilia assembly? Are their expression also directly linked to ciliogenesis in other types of cells?

Thank you for the thoughtful questions. The ubiquitous expression of XAP5 in mice can be understood in light of its role as an ancient transcription factor for cilia and flagellar assembly. Given that cilia are present on nearly every cell type in the mammalian body (O'Connor et al., 2013), this broad expression pattern makes sense. In fact, XAP5 serves not only as a master regulator of ciliogenesis but also as a critical regulator of various developmental processes (Kim et al., 2018; Lee et al., 2020; Xie et al., 2023).

Our current unpublished work demonstrates that XAP5 is essential for primary cilia assembly in different cell lines. The loss of XAP5 protein results in abnormal ciliogenesis, further supporting its vital role in ciliary formation across different cell types.

We believe that the widespread expression of XAP5 reflects its fundamental importance in multiple cellular processes, including ciliogenesis, development, and potentially other cellular functions yet to be discovered.

(4) XAP5L causes impairs flagellar assembly. Have the authors observed any other physiological defects in the absence of XAP5L in mouse models? Such as hydrocephalus and/or tracheal defects?

Thank you for the questions. We have carefully examined *XAP5L* KO mice for other physiological defects. To date, we have not observed any additional physiological abnormalities. Specifically, we assessed the condition of tracheal cilia in *XAP5L* KO mice and found no significant differences compared to wild-type (WT) mice, as illustrated in Author response image 1 below.

References

Jung, M., Wells, D., Rusch, J., Ahmad, S., Marchini, J., Myers, S. R., & Conrad, D. F. (2019). Unified single-cell analysis of testis gene regulation and pathology in five mouse strains. *Elife, 8*. doi:10.7554/eLife.43966

Kaya-Okur, H. S., Wu, S. J., Codomo, C. A., Pledger, E. S., Bryson, T. D., Henikoff, J. G., . . . Henikoff, S. (2019). CUT&Tag for efficient epigenomic profiling of small samples and single cells. *Nat Commun, 10*(1), 1930. doi:10.1038/s41467-019-09982-5

Kim, Y., Hur, S. W., Jeong, B. C., Oh, S. H., Hwang, Y. C., Kim, S. H., & Koh, J. T. (2018). The Fam50a positively regulates ameloblast differentiation via interacting with Runx2. *J Cell Physiol, 233*(2), 1512-1522. doi:10.1002/jcp.26038

Lee, Y.-R., Khan, K., Armfield-Uhas, K., Srikanth, S., Thompson, N. A., Pardo, M., . . . Schwartz, C. E. (2020). Mutations in FAM50A suggest that Armfield XLID syndrome is a spliceosomopathy. *Nature Communications, 11*(1). doi:10.1038/s41467-020-17452-6

O'Connor, A. K., Malarkey, E. B., Berbari, N. F., Croyle, M. J., Haycraft, C. J., Bell, P. D., . . . Yoder, B. K. (2013). An inducible CiliaGFP mouse model for in vivo visualization and analysis of cilia in live tissue. *Cilia, 2*(1), 8. doi:10.1186/2046-2530-2-8

San Agustin, J. T., Pazour, G. J., & Witman, G. B. (2015). Intraflagellar transport is essential for mammalian spermiogenesis but is absent in mature sperm. *Mol Biol Cell, 26*(24), 4358-4372. doi:10.1091/mbc.E15-08-0578

Xie, X., Li, L., Tao, S., Chen, M., Fei, L., Yang, Q., . . . Chen, L. (2023). Proto-Oncogene FAM50A Can Regulate the Immune Microenvironment and Development of Hepatocellular Carcinoma In Vitro and In Vivo. *Int J Mol Sci, 24*(4). doi:10.3390/ijms24043217